# Quality of Spontaneous Reports of Adverse Drug Reactions Sent to a Regional Pharmacovigilance Unit

**DOI:** 10.3390/ijerph19073754

**Published:** 2022-03-22

**Authors:** Mário Rui Salvador, Cristina Monteiro, Luísa Pereira, Ana Paula Duarte

**Affiliations:** 1Public Health Unit, Local Health Unit of Guarda, 6301-858 Guarda, Portugal; 2Pharmacovigilance Unit of Beira Interior, Faculty of Health Sciences, University of Beira Interior, 6200-506 Covilha, Portugal; ufarmabi@fcsaude.ubi.pt (C.M.); apcd@ubi.pt (A.P.D.); 3Department of Mathematics, University of Beira Interior, 6201-001 Covilha, Portugal; lpereira@ubi.pt; 4The Health Science Research Centre, University of Beira Interior, 6200-506 Covilha, Portugal

**Keywords:** adverse drug reactions, pharmacovigilance, spontaneous report, quality, public health

## Abstract

Spontaneous reports (SRs) of adverse drug reactions (ADRs) remain the basis of pharmacovigilance systems. The main objective of this study was to evaluate the quality of SRs received by the Pharmacovigilance Unit of Beira Interior, in Central Portugal. The second objective was to identify factors associated with complete SRs. SRs received between 1 January 2017 and 31 October 2019 were analyzed. SR information was classified as “mandatory” or “recommended” criteria. SR were then grouped into three categories (well, slightly, and poorly documented). Association between “well documented” SR and confounding variables was estimated using a multiple logistic regression model. The results showed 22.4% of SRs are “well documented”, and 41.2% are “poorly documented”. Most of the complete SRs correspond to non-serious ADRs (55.8%), with a negative association between complete SRs and serious ADRs (OR = 0.595, [95% CI 0.362–0.977], *p* = 0.040). There is also a significant association between complete SRs and e-mail notification (OR = 1.876, [95% CI 1.060–3.321], *p* = 0.002). The results highlight the small amount of SR documentation sent to pharmacovigilance systems. There is an association between non-serious ADRs and complete SRs. These results reinforce the need for training for notification of ADRs and that these SRs include as much information as possible for an effective drug risk management.

## 1. Introduction

An adverse drug reaction (ADR) is the “appreciably harmful or unpleasant reaction resulting from an intervention related to the use of a medicinal product, which provides risks of future administration and justifies prevention or specific treatment or change in the therapeutic regimen or withdrawal of the product”. An ADR can be classified as either serious (if it causes death, life-threatening illness, hospitalization, disability/incapacity, or a congenital anomaly, or is considered clinically important by the notifier or expert), or non-serious [1].

An ADR is an adverse event in which a causal relationship between a medicinal product and an occurrence is suspected. However, despite the definitional differences between an ADR and an adverse event, for regulatory reporting purposes, if an event is spontaneously reported, even if the relationship is unknown or unstated by the healthcare professional or consumer as the primary source, it meets the definition of an adverse reaction [2].

It is estimated that, in the European Union, ADRs account for 5% of total hospital admissions, that 5% of hospitalized patients will suffer an ADR during hospitalization, and that ADRs cause 197,000 deaths annually [1]. In the United States of America, ADRs are the fifth leading cause of death [3]. In terms of costs, ADRs are estimated to represent 15 to 20% of total hospital costs, to extend hospitalizations by 4 days and to, annually, cost USD 6 million in a 700-bed hospital [4]. Thus, morbidity and mortality associated with drug use and its socio-economic transcendence make ADRs an important public health problem.

Pharmacovigilance was defined by the World Health Organization (WHO) as the science and activities related to the detection, evaluation, understanding, and prevention of adverse effects or any other drug-related safety problem [5]. Pharmacovigilance standards are therefore necessary to protect public health in order to prevent, detect, and assess adverse reactions of medicinal products placed on the market, since only after their introduction to the market is it possible to fully understand their safety profile [6]. Among the different pharmacovigilance methodologies, SRs of ADRs sent to pharmacovigilance units remain the basis of pharmacovigilance systems [7]. Spontaneous reports allow the assessment and management of drug-related risks and are one of the main sources of decision making for drug withdrawals from the market [8]. Its advantages are its low cost and ease of implementation [7].

However, the SR system has some disadvantages, namely, under-reporting and incomplete provision of data [7]. It is estimated that, in clinical practice, less than 5% of ADRs are notified even in contexts where notification is mandatory [9]. According to data from the WHO Uppsala Monitoring Centre (UMC), in 2014, only 13% of SRs had a good degree of information completion [7]. Regulatory agencies have identified the need for quality management systems as an essential element of good pharmacovigilance practices. The degree of completion of an SR is one of the quality parameters that should be considered [8]. According to Good Pharmacovigilance Practices (GVPs), an SR of an ADR is considered valid if it includes: (1) an identifiable notifier; (2) an identifiable user (characterized by initials, date of birth, gender, or age); (3) one or more suspicious drugs; and (4) one or more suspected ADRs [10]. In addition to these mandatory data, a well-documented SR should also contain information on basic medical condition, comorbidities, concomitant medication, clinical evolution of the patient, therapy implemented to treat the ADR, complementary means of diagnosis, and information on the response to suspension and reintroduction of the drug [8,11,12].

In Portugal, SRs can be notified to regional pharmacovigilance units (RPUs) in four ways: (1) an online form of the ADR Portal of the National Medicines and Health Products Authority-INFARMED, IP (Infarmed); (2) a paper notification form; (3) by e-mail; (4) by telephone [5]. RPUs collect and process all SRs of ADRs notified in the coverage region. All reports are individually evaluated, and additional information is required from the notifier when needed. The causality imputation is carried out by medical experts through the method of global introspection [13]. Once evaluated, each report is submitted to the Portuguese national database, the Infarmed ADR Portal [5]. The Pharmacovigilance Unit of Beira Interior (PUBI) is one of the nine RPUs that exist in Portugal. It is headquartered in the Faculty of Health Sciences of the University of Beira Interior, Covilhã, and covers the districts of Castelo Branco, Guarda, and Viseu, in the Central Region of Portugal, with a total of approximately 735,000 inhabitants [5].

The main objective of this study was to evaluate the quality, measured by the completeness of the notification criteria, of the SRs of ADRs received by the PUBI. The second objective was to identify factors associated with SRs that were considered to be complete.

## 2. Materials and Methods

To perform this study, all SRs of ADRs received by the PUBI between 1 January 2017 and 31 October 2019 were selected and analyzed.

As stated before, for regulatory reporting purposes, a spontaneously reported event, even if the relationship is unknown or unstated by the healthcare professional or consumer as a primary source, meets the definition of an adverse reaction [2]. As all SRs sent to the PUBI have at least one suspected drug declared by the notifier, according to the GVPs, all adverse events reported to PUBI were considered to be ADRs.

This was an observational study. All data related to ADRs were voluntarily sent by the notifier to PUBI. All data were anonymized at the source. The authors did not have access to any personal information or clinical file that could identify the patient. Identification of the patients was not possible and, therefore, ethico-legal principles were fulfilled. All forms of notification were considered, namely via ADR portal, by e-mail, by telephone, and by mail. Received reports that were considered null due to the lack of information necessary for their validation were excluded from the analysis.

Based on the literature [8,10,11], in this study, the data collected were classified as “mandatory” or “recommended” criteria. “Mandatory” criteria included: (1) name initials of the patient; (2) complete date of birth of the patient; (3) sex of the patient; (4) date of occurrence of ADR; (5) date of administration of the suspect medicine(s). “Recommended” criteria included: (1) brief medical history of the patient; (2) concomitant medication; (3) clinical evolution; (4) documentation of the diagnosis of the reaction and/or results of medical examination and/or results of complementary means of diagnosis and therapy (CMDT).

According to previous literature [8] and according to the experience of the pharmacovigilance professionals of the PUBI, SR were grouped into three categories constructed according to the presence or absence of “mandatory” and “recommended” criteria:“Well documented”: if the five “mandatory” criteria are met and three of the four “recommended” criteria are met;“Slightly documented”: if five “mandatory” criteria are met and at least one “recommended” criterion is met;“Poorly documented”: all other situations.

The association between “well documented” SR and independent variables (ADR seriousness, patient age group, notification mode, notifier type) was estimated using a multiple logistic regression model. To perform this analysis, the three categories of SR were regrouped into two classes: “complete” (all “well documented” SR) and “incomplete” (all “slightly documented” and “poorly documented” SR). Three age groups were constructed according to the age of the patients: “young”, when under 18 years; “adults” when aged between 18 and 65 years; “elderly”, when over 65 years. The association measure considered was the odds ratio, calculated using the chi-square test. Statistically significant associations were considered those with *p*-value < 0.05. Statistical analysis was performed using the IBP SPSS Statistics version 24 software.

## 3. Results

### 3.1. Characterization of Spontaneous Reports of Adverse Drug Reactions

Between 1 January 2017 and 31 October 2019, a total of 425 SRs were notified to PUBI, of which 238 (56.0%) were considered “serious”.

The notifications involved more women than men (62.6 vs. 37.4%, omitted data: 8.2%). The mean age (±SD) of patients at the time of the ADR was 50.8 ± 24.2 years (range 0–95 years) and most notifications were related to adults and elderly (55.1 and 33.3% respectively, omitted data: 4.0%).

The main forms of notification were via ADR portal and via e-mail (42.8 and 33.4%, respectively). The notifications came mainly from doctors and patients (55.8 and 21.6%, respectively).

### 3.2. Quality of Spontaneous Reports

#### 3.2.1. Completeness of Spontaneous Notifications

Table 1 shows the fulfilling percentage of mandatory and recommended criteria of the SR, distributed among the three built categories.

The criteria most frequently present in the notifications received by the PUBI were “patient initials” (present in 98.4% of SRs), “clinical evolution” (in 96.0% of SRs), and “patient sex” (in 91.8% of SRs). The least frequently met criteria were “documentation/CMDT results” (in 6.8% of SRs), “concomitant medication” (in 34.8%) and “date of birth” (in 72.0%).

According to the three defined categories, the most frequent notifications were “poorly documented” (41.2%), followed by “slightly documented” notifications (36.5%). “Well documented” notifications accounted for 22.4% of the total. In the category of “well documented” notifications, all criteria were fully met except “concomitant medication” (89.5%) and “CMDT documentation/results” (16.8%). In the category of “slightly documented” notifications, all recommended criteria presented missing information, and the criteria less frequently complete were “concomitant medication” (7.7%) and “CMDT documentation/results” (3.9%). In the category of “poorly documented” notifications, among the mandatory criteria, missing information was more frequent in the criteria “date of administration of the drug” (65.1%) and “patient’s date of birth” (32.0%) and, among the recommended criteria, missing information was more frequent in the criteria “concomitant medication” (29.1%) and “MCDT documentation/results” (4.0%).

#### 3.2.2. Factors Associated with Complete Spontaneous Notifications

Table 2 presents the results of the univariate and multivariate analysis of the association between complete and incomplete SRs and the independent variables “year of notification”, “seriousness of ADR”, “age group of the individual who suffered ADR”, “SR mode”, and “type of notifier”.

Of the total of 425 SRs received by PUBI, 95 (22.4%) were considered complete and 330 (77.6%) were considered incomplete. The year 2018 had the highest occurrence of SRs (44.0%) and incomplete SRs (47.0%), and 2019 recorded the highest percentage of complete SRs (36.8%). It should be noted that the study includes all notifications of the years 2017 and 2018, but in 2019 only SRs sent until 31 October 2019 were included. The majority of complete SRs were related to non-severe ADRs (55.8%), whereas most incomplete SRs corresponded to serious ADRs (59.4%). In both the complete SR group and the incomplete SR group, the majority of SRs were related to adults (58.5 and 54.1%, respectively), followed by notifications related to elderly (35.1 and 32.8%, respectively) and young people (6.40 and 13.1%, respectively). Regarding the notification mode, it was verified that 42.8% of the total SRs were made via the ADR Portal. It was also verified that 44.5% of the incomplete SRs were notified through the ADR Portal whereas, among the complete SRs, 51.6% were notified by e-mail. Most of the SRs were performed by physicians (55.8%), and the same was observed in the complete SRs (54.7%) and incomplete SRs (56.1%).

The univariate analysis allowed identification of a statistically significant association between complete SRs and “year of notification” (*p* = 0.08), “serious ADR” (*p* = 0.09) and “notification mode” (*p* < 0.001), and did not identify a statistically significant association with “age group” (*p* = 0.220) and “type of notifier” (*p* = 0.540).

The multivariate analysis allowed the identification of a negative association between complete SRs and serious ADRs (OR = 0.595, [95% CI 0.362–0.977], *p* = 0.040) and a positive association between complete SRs and e-mail notification (OR = 1.876, [95% CI 1.060–3.321], *p* = 0.002).

## 4. Discussion

The study showed that only 22.4% of the notifications sent to PUBI are “well documented” (or “complete SR”), whereas 41.2% are “poorly documented”. The criteria that registered the greatest lack of information were “concomitant medication” (65.2%) and “CMDT documentation/results” (93.2%).

In addition, the study found that the majority of complete SRs correspond to non-serious ADRs (55.8%), whereas most incomplete SRs are related to serious ADRs (59.4%); thus, a negative association was observed between complete SRs and serious ADRs (OR = 0.595, [95% CI 0.362–0.977], *p* = 0.040). This means that the probability of a serious ADR being reported with a lack of information is higher than the probability of a non-serious ADR being reported with a lack of information. There was also a positive association between complete SRs and e-mail notification (OR = 1.876, [95% CI 1.060–3.321], *p* = 0.002). No significant association was found with the variable “notifier type”.

Different studies have been conducted to assess the quality of reports of adverse drug reactions [7,8,12,14,15]. The results of this study are in line with the literature regarding the overall quality of adverse reaction notifications [8,15,16,17], but are contrary to the literature regarding the association between the degree of completeness of SRs and the seriousness of ADRs [8].

A study conducted in France in 2016 assessed the quality of notifications sent by general practitioners to a regional pharmacovigilance unit (*n* = 613), measured by the level of mandatory and non-mandatory criteria, classifying them as “well documented”, “slightly documented”, and “poorly documented”. The results showed that only 12.7% of the notifications were “well documented”, whereas 68.5% were considered “slightly documented” and 18.8% “poorly documented” [8]. The methodology used was very similar to that of the present study and so were the results. Another study conducted in 2013 by the UMC used an evaluation score (the vigiGrade completeness score, C) to measure the amount of clinically relevant information present in the global notifications recorded in the WHO VigiBase, and concluded that only 13.0% (*n* = 3.3 million) of the notifications sent between 2007 and 2012 were considered “well documented” [17]. In Spain, a study developed in 2016 used the same approach in the evaluation of notifications received by the Regional Centre of Catalonia throughout 2014 (*n* = 350), and concluded that 42.5% of the notifications were “well documented”, with differences between health professionals (54.7%) and pharmaceutical industry (23.4%) [15]. In China, in 2018, the quality of SRs notified to a regional pharmacovigilance center between January 2015 and December 2017 (*n* = 1139) were assessed through a specific quality assessment system based on statistical analysis. The results showed that only 1.4% of the SRs and only 8.3% of the SRs of severe ADRs were classified as good quality [7]. In Mexico, researchers assessed the quality of notifications sent to the national pharmacovigilance system in 2007 (*n* = 370) and 2008 (*n* = 371), according to national guidelines, and it was concluded that 32.0 and 40.0% of the notifications received each year had incomplete information [16].

Regarding the degree of completion of the information on each field of the SR, the results obtained are in line with the literature [8]. In a study conducted in 2017 in Brazil, it was found that 40.9% (*n* = 1000) of SRs recorded information regarding “concomitant diseases”. The previously mentioned study undertaken in France in 2016 identified 30.7% and 27.7% of SRs with information regarding “concomitant medication” and “documentation/results of MCDT”, respectively, whereas the study conducted in Spain in 2016, also previously mentioned, concluded there was an almost absence of information on “date of ADR” (11.5%), “medicine administration date” (14.3%), and “clinical evolution” (16.5%).

Taking account of the variability in the methodologies of the different studies, all highlight the small amount of notification documentation sent to pharmacovigilance systems, compromising quality and signal management activities.

The GVPs recommend that a well-documented notification should include the description of concomitant medication and complementary means that document the etiology of reaction [10].

Information on concomitant medication is essential in the causality imputation in order to exclude the relationship of the described reaction with other drugs and thus to allow establishing a causal link between the reaction and the suspect drug. It is also important in the evaluation of potential drug interaction [5]. Knowledge of the “date of ADR” and the “medicine administration date” is fundamental to establish the period that measures exposure and the suspected effect, and to be able to infer, from its temporal plausibility, a fundamental criterion in the WHO classification of the causality of ADRs [5].

The results of this study allow us to conclude that there is a need to harmonize the required and mandatory information to report ADRs. Among countries, there are discrepancies that skew the interpretation and comparability of data at the European level. For example, there are systems that do not even contemplate the possibility of documenting ADRs with CMDT results [9].

Regarding the association between the severity of ADRs and the degree of completeness of SRs, the study previously conducted in France in 2016 identified a positive association between complete SRs and serious ADRs (OR = 1.70 [CI 95% 1.04–2.76], *p* = 0.03) [8].

However, the present study identified an inverse association between the complete SRs sent to the UFBI and the serious ADR variable, with OR = 0.595 ([95% CI 0.362–0.977], *p* = 0.040). Thus, it was found that most severe ADRs result in incomplete SRs (44.2% complete SR vs. 59.4% incomplete SR) and that serious ADRs are most likely reported in an incomplete way, compared to a non-serious ADR.

Due to their health consequences, serious ADRs are those that represent the greatest risk for drug users. As evidenced by this study, the higher probability of a severe ADR being reported in an incomplete manner, when compared to a non-serious ADR, reinforces the need for training about notification and indicates that as much information as possible is needed for a causality-based imputation.

It is certain that underreporting of ADRs persists and that this is partly due to the excessive work of health professionals [18,19]. It is therefore necessary to counterbalance the requirement to fill in more notification fields with the concern of not requiring too much time and effort from notifiers, which can undermine their motivation in reporting adverse reactions.

Moreover, the positive association observed in this study between complete SRs and e-mail notification (OR = 1.876, [CI 95% 1.060–3.321], *p* = 0.002), having the reference “Notification via ADR Portal”, may mean a need to reformulate the fields that must be completed for a SR made in the ADR Portal, thus providing a greater quantity of mandatory information for the validation of an SR, particularly if the ADR is serious. In addition, this association of a complete SR with the notification mode may also mean the ADR Portal should be promoted as the preferred route of notification of ADRs, both to users and professionals.

### Study Strengths and Limitations

As far as we could verify, this is the first study in Portugal that examined the quality of SRs sent to a regional pharmacovigilance unit, measured by the degree of completion of fields of an SR and its association with the seriousness of an ADR, the type of notifier, or the mode of notification. In addition to the innovative factor, all notifications received between 1 January 2017 and 31 October 2019 were considered, and sampling methods were not used.

However, the study is not exempt from limitations. The construction of the “well documented” category, without the need to meet all the recommended criteria identified, may have affected the results. In addition, the definition of “mandatory” and “recommended” criteria, although based on GVPs, depended, in part, on the subjective understanding of pharmacovigilance professionals. Moreover, the selection of independent variables to be included in the multivariate analysis may not include all those that allowed a better adjustment for confounding.

## 5. Conclusions

The most important conclusions of this study are, first, the evidence of the low level of criteria documentation needed to make a reasoned causality imputation of spontaneous reports of adverse drug reactions and, second, the high proportion of serious adverse drug reactions reported through incomplete spontaneous reports and, thus, the higher probability of a serious adverse drug reaction being reported incompletely compared to a non-serious adverse drug reaction. This undermines a well-founded causality imputation and, thus, the effective risk management of the use of medicines. The results reinforce the need for training about notification, and that as much information as possible is provided for a causality-based imputation, especially when a serious ADR is suspected.

## Figures and Tables

**Table 1 ijerph-19-03754-t001:** Proportions of the nine mandatory and recommended criteria of spontaneous reports sent to the Pharmacovigilance Unit of Beira Interior between 1 January 2017 and 31 October 2019, distributed in three categories (“well documented”, “slightly documented”, “poorly documented”).

	Well Documented *n* (%)	Slightly Documented *n* (%)	Poorly Documented *n* (%)	Total *n* (%)
Mandatory criteria				
Patient initials	95 (100)	155 (100)	168 (96.0)	418 (98.4)
Patient’s date of birth	95 (100)	155 (100)	56 (32.0)	306 (72.0)
Patient sex	95 (100)	155 (100)	140 (80.0)	390 (91.8)
Date of ADR	95 (100)	155 (100)	121 (69.1)	371 (87.3)
Medicine administration date	95 (100)	155 (100)	114 (65.1)	364 (85.6)
Recommended criteria				
Medical history	95 (100)	102 (65.8)	121 (69.1)	318 (74.8)
Concomitant medication	85 (89.5)	12 (7.7)	51 (29.1)	148 (34.8)
Clinical evolution	95 (100)	153 (98.7)	160 (91.4)	408 (96.0)
CMDT documentation/results	16 (16.8)	6 (3.9)	7 (4.0)	29 (6.8)
SR (*n* = 425)	95 (22.4)	155 (36.5)	175 (41.2)	

Note: ADR, adverse drug reaction; CMDT, complementary means of diagnosis and therapy; SR, spontaneous report.

**Table 2 ijerph-19-03754-t002:** Results of the univariate and multivariate analysis of the study of the association between complete SRs and the variables “year of notification”, “serious ADR”, “age group”, “notification mode”, and “type of notifier”.

	Complete SR *n* (%)	Incomplete SR *n* (%)	Total	Univariate Analysis			Multivariate Analysis		
				*p*	OR	CI 95%	*p*	OR	CI 95%
Year of SR				0.008			0.021		
2017	28 (29.5)	53 (16.1)	81 (19.1)		Ref			Ref	
2018	32 (33.7)	155 (47.0)	187 (44.0)		0.391	(0.215–0.709)		0.391	(0.201–0.762)
2019 (until 31 October)	35 (36.8)	122 (37.0)	157 (36.9)		0.547	(0.300–0.982)		0.575	(0.287–1.155)
Serious ADR				0.009			0.04		
No	53 (55.8)	134 (40.6)	187 (44.0)		Ref			Ref	
Yes	42 (44.2)	196 (59.4)	238 (56.0)		0.542	(0.342–0.859)		0.595	(0.362–0.977)
Age group				0.22			NS	-	-
Young	6 (6.40)	41 (13.1)	47 (11.5)		Ref				
Adult	55 (58.5)	170 (54.1)	225 (55.1)		2.211	(0.891–5.487)			
Elderly	33 (35.1)	103 (32.8)	136 (33.3)		2.189	(0.853–5.617)			
Mode of SR				<0.001			0.002		
ADR Portal	35 (36.8)	147 (44.5)	183 (42.8)		Ref			Ref	
E-mail	49 (51.6)	93 (28.2)	142 (33.4)		2.213	(1.335–3.669)		1.876	(1.060–3.321)
Telephone	9 (9.5)	79 (23.9)	88 (20.7)		0.478	(0.219–1.046)		0.497	(0.197–1.012)
Mail	2 (2.1)	11 (3.3)	13 (3.1)		0.764	(0.162–3.602)		0.331	(0.041–2.697)
Notifier				0.54			NS	-	-
Patient	17 (17.9)	75 (22.7)	92 (21.6)		Ref				
Doctor	52 (54.7)	185 (56.1)	237 (55.8)		1.24	(0.674–2.282)			
Pharmacist	18 (18.9)	46 (13.9)	64 (15.1)		1.726	(0.809–3.683)			
Other professional	8 (8.4)	24 (7.3)	32 (8.5)		1.471	(0.564–3.832)			
Total	95 (22.4)	330 (77.6)	425 (100.0)						

Note: ADR, adverse drug reaction; CI, confidence interval; NS, non-significant; OR, odds ratio; Ref, reference; SR, spontaneous report.

## Data Availability

All data are available at Pharmacovigilance Unit of Beira Interior archive.

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
