# Peer review of "Quality of Spontaneous Reports of Adverse Drug Reactions Sent to a Regional Pharmacovigilance Unit"

_ijerph, 2022, doi:10.3390/ijerph19073754_

Round 1
Reviewer 1 Report
Major comments
- A sentence as a paragraph in this way of writing is irregular and non-standard.
- The authors did not analyze which criteria influenced ADR, such as‘Mandatory' or ‘Recommended'.
- It is suggested that the correlation between SR and ADR can be expressed more intuitively by
- Please standardize references, such as line 77, 226, 222, 208.
- Please revise “Identification of the patient were”.
Author Response
Dear Reviewer
Thank you very much for your comments and suggestions.
I'm sending you a document with the notes/answers to your comments.
Best regards,
Mario Rui Salvador

Reviewer 2 Report
Strengths:
- Well written Introduction
- Clear description of methods
- Balanced discussion
Weaknesses:
- Abstract: I am a little bit confused because of these two contradictory statements:
- Line 19: “statistically significant association between complete SR and serious ADRs”
- Line 22: “There is an association between non-serious ADR and complete SR”
- A time range covering the “Covid era” would be of interest, especially considering the frequency of reporting related to the vaccines. However, I assume this may be too demanding.
- Line 97: I am a little bit confused about the Mandatory criteria. The mandatory criteria included complete date of birth and dates of occurrence but not the drug or the description of adverse reaction. Maybe the documentation of the suspected drug is covered by point (5) (Line 98-99), but this is not clear to me. Why is, for example, the day of the month of the DOB so relevant? I would argue that this is a negligible information. Previous studies used a similar definition, which you also referenced. However, is there an explanation for this? I don’t want to stress this too much, but a perfectly documented case report, which only provides month and year of birth, would be classified as poorly documented. Physician may even intentionally omit the DOB to guarantee the anonymity of the patients. Considering Table 1, the DOB was the main problem of poorly documented SRs (only 32% had it). This should at least be discussed.
- Please provide a definition of ‘serious’ ADRs.
- The term ‘clinical evolution’ is a little bit unclear. What do you mean exactly?
- Line 152-153: These numbers refer to the total sample and not to the ‘poorly documented’ group? (according to the columns of Tab 1)
- Table 2: The OR for the years 2018 and 2019 are below 1, but the numbers of SRs is higher than in 2017 (ie the reference). Shouldn’t be the OR be >1 in this case? (Same is true for Seriousness)
- The wording positive or negative/inverse correlation of two binary variables (eg. Line 265 and others) is rather unconventional and a little bit confusing throughout the manuscript.
- The manuscript is overall well written. However, Lines 303-306 are the ‘most import conclusion’ of this manuscript, and yet this sentence is difficult to understand. Consider rephrasing this section.
- I think the conclusions can be improved. What is really the key message of this work? What can we learn? What do these data add to the current knowledge and is there a need for action?
Author Response

(The authors gave the same response as above.)

Reviewer 3 Report
The manuscript presents an interesting work regarding how pharmacovigilance systems can collect information from reports. The work is generally well presented and organized but there are some issues that overal mark the manuscript with major/minor revisions. Although the most of them can be adressed my major concern that marks the manuscript mith major comments is the adverse drug event/adverse drug reactions definition that is not so clear in the text. Adverse drug events, adverse drug reactions and side effects are three different cases in clinical practice. Some comments follow:
1) Usually there is not an adverse drug reaction that is reported but an adverse drug event. The pharmacovigilance system will provide feedback if the event is a reaction (attributed to the pharmacological active compound) or event (attributed to the formulation).
2) Please state even in abstract that the study refers to Portugal.
3) Mandadory-recommended under what criteria?
4) Under what terms is it an ADR serious? Did the authors see any based on dose or not (Rawlins & Thompson 1977) or based on Edwards & Aronsosn, 2000 ABCDEFG categories or Siegel & Hartwig (1992) Level 1-7?
5) How this work is related with Drug Saf. 2017 Oct;40(10):855-869. doi: 10.1007/s40264-017-0572-8. and Clin Pharmacol Ther
. 2020 Mar;107(3):521-529. doi: 10.1002/cpt.1678. After all the study is taking place within EU.
Author Response

(The authors gave the same response as above.)

Round 2
Reviewer 2 Report
Thank you for your corrections. No further comments form my side.
Author Response
Dear Reviewer
Thank you very much for your comments and suggestions.
Best regards,
Mario Rui Salvador
Reviewer 3 Report
The authors presented an updated version of their work and addressed the comments made from the reviewers.
I still believe that the proper to state is that SRs refer to adverse drug events (ADEs) which create pharmacovigilance signals that are analyzed, creating feedbacks and updating SPCs as ADRs. ADEs can be from drug administered or other factors such as comorbidities, compliance, genetics, environmental reasons, other drugs and interactions, diet habits etc. So, best to my knowledge an ADR is an ADE with causal relationship with the medication. The authors agree with that. For example the Novartis vaccine-induced VITT which was an ADE that causally associated with the covid-19 vaccine and now is an ADR.
So maybe this work is more with ADE and less with ADRs. If I am wrong and the study included only ADRs (somehow evaluated) that are described in the SPCs then my apologies but usually in yellow cards or periodical report systems etc. for pharmacovigilance usually an adverse event is stated. So it may be proper, if the description ADE was used throughout the manuscript. However, the authors were based their work with PVC description so they make the assumption that the SRs are evaluated as drug-related ADEs. So maybe they should include a paragraph stating-describing ADEs and ADRs in discussion in my opinion.
Apart of that the manuscript can be processed but it needs a heavy edit regarding its structure, too many paragraphs etc. Also conclusion can be of a paragraph.
Author Response

(The authors gave the same response as above.)
